# Isotope sample preparation of diatoms for paleoenvironmental research

**George E. A. Swann** *, **Andrea M. Snelling**

School of Geography, University of Nottingham, Nottingham, United Kingdom

* george.swann@nottingham.ac.uk

**Data Availability Statement:** All relevant data are within the paper and its Supporting Information files.

**Funding:** This study was funded by Natural Environment Research Council (https://www.ukri.

## Abstract

Isotopes in diatoms are increasingly used in palaeoenvironmental studies in both lacustrine and marine settings, enabling the reconstruction of a range of variables including temperature, precipitation, salinity, glacial discharge, carbon dynamics and biogeochemical cycling. This protocol details an optimised methodology for extracting diatoms for isotope analysis from sediment samples, using a range of chemical and density separation techniques that minimise sample loss and avoids the need for expensive equipment. Whilst designed for the extraction of diatoms for oxygen, silicon and carbon isotope analysis, additional stages are outlined for the analysis of other isotopes that are of increasing interest to the palaeo community (e.g., boron and zinc). The protocol also includes procedures for assessing sample purity, to ensure that analysed samples produce robust palaeoenvironmental reconstruction. Overall, the method aims to improve the quality of palaeoenvironmental research derived from isotopes in diatoms by maximising sample purity and the efficiency of the extraction process.

## Introduction

Isotopes in diatoms (e.g., $\delta^{13}C$, $\delta^{15}N$, $\delta^{18}O$, $\delta^{30}Si$) provide a key source of palaeoenvironmental information in marine and lacustrine environments where carbonates not readily preserved in the sedimentary environment [1, 2]. Whilst the emergence of isotopes in diatoms as a palaeoenvironmental proxy has occurred alongside the development of mass-spectrometry techniques for their analysis [3–10], projects are often hindered by difficulties in extracting sufficient diatoms for analysis without the presence of non-diatom contaminants. Here we describe a protocol, suitable for Masters and PhD students, that has been used at the University of Nottingham for over a decade to obtain pure diatom samples from a sediment matrix, before samples are analysed at the National Environmental Isotope Facility (British Geological Survey) for $\delta^{13}C$, $\delta^{18}O$ and $\delta^{30}Si$ [6, 9]. Extensions/deviations to the core methodology are also outlined for samples that will be analysed for other/novel isotope systems that are of increasing interest to the palaeo community such as $\delta^{11}B$ [11] and $\delta^{66}Zn$ [12]. This protocol is not fully compatible with accepted diatom protocols for $\delta^{15}N$. Instead, samples for diatom $\delta^{15}N$ should be prepared following [5, 13]. Caution should also be exerted when applying this, or indeed any, protocol to living/cultured diatom frustules due to the potential for post-mortem oxygen isotope exchange [14].

Typically, in our experience, only 50–70% of sediment samples can be sufficiently cleaned to remove non-diatom contaminants and generate enough material for isotope analysis. This

org/councils/nerc/) grants NE/F012969/1, NE/F012969/2, NE/I005889/1 and NE/G004137/1 to GS. The funders had and will not have a role in study design, data collection and analysis, decision to publish, or preparation of the manuscript.

**Competing interests:** The authors have declared that no competing interests exist.

success rate varies between sites and different aged samples and is predominantly determined by the amount of raw material available, the concentration of diatoms and the presence/abundance of other types of biogenic silica (e.g., siliceous sponges and radiolaria) which can be problematic to separate. The protocol presented here has been tested on a wide variety of different aged lacustrine and marine sediments (0–3.4 Ma) and optimised to minimise the risk of sample loss. It also outlines stages for assessing and quantifying sample purity, as well as for checking that the extracted diatoms are not contaminated by diagenesis, dissolution or other processes that might have altered the isotopic signature.

## Materials and methods

The protocol described in this peer-reviewed article is published on protocols.io, https://dx.doi.org/10.17504/protocols.io.36wgq4knovk5/v2 and is included for printing as S1 File with this article.

## Expected results

Using this protocol, we have been able to obtain pure diatom samples for isotope analysis with minimal material loss (Fig 1). The procedure has been successfully used on raw sediment

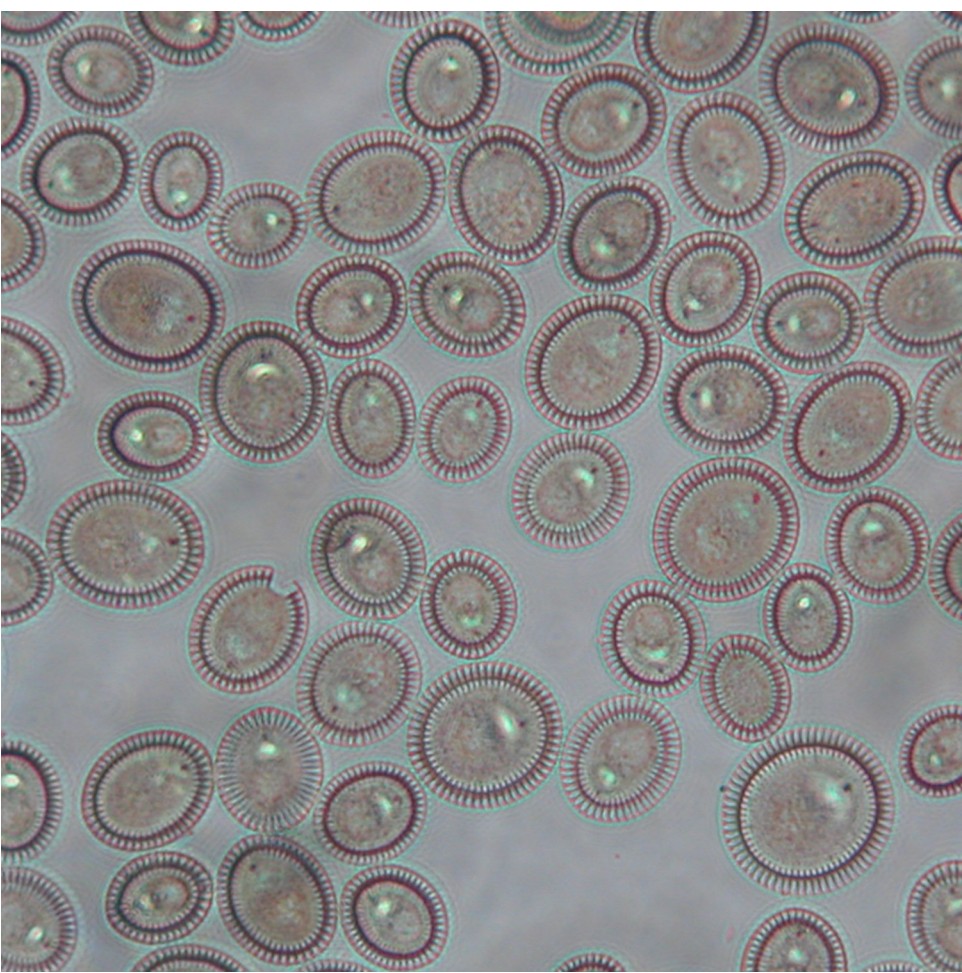

**Fig 1. Example of a purified diatom sample (under light microscope) following the use of this protocol.** Sample from Lake El'gygytgyn, Russia [15].

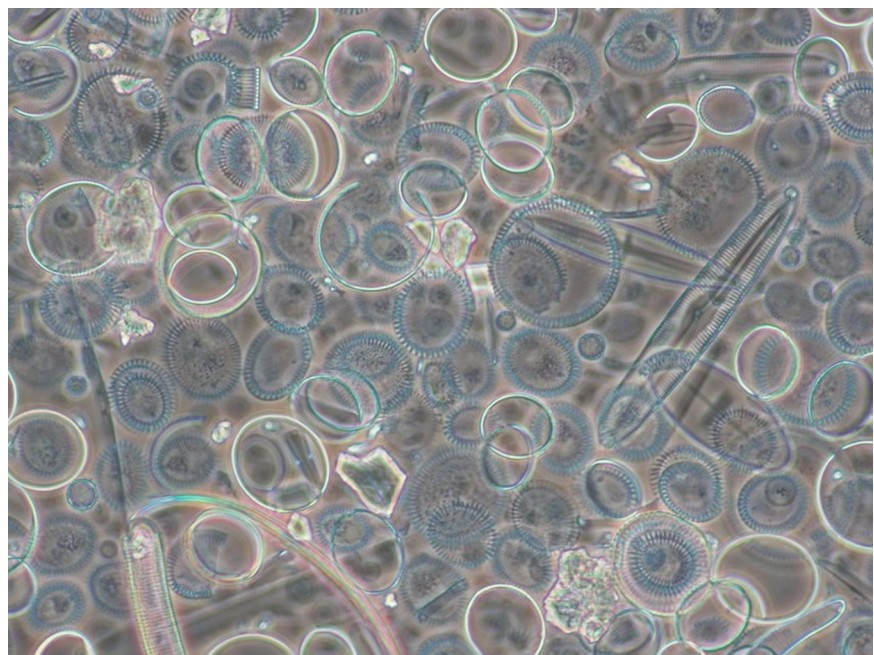

**Fig 2. Example of a diatom sample (under light microscope) contaminated with aluminosilicates.** Sample from Lake El'gygytgyn, Russia [15].

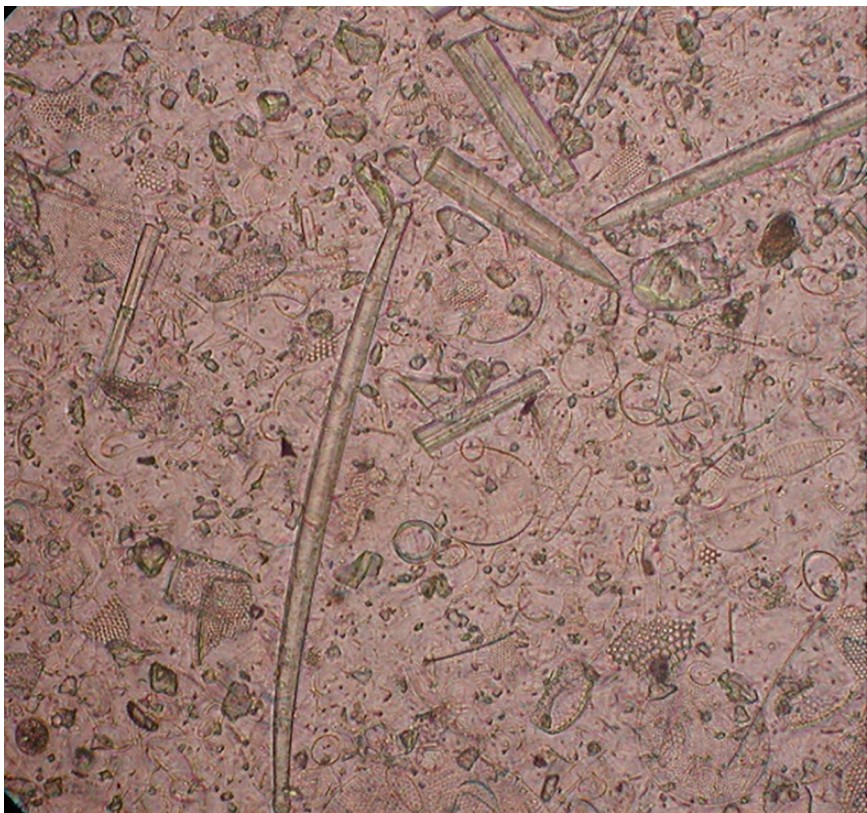

**Fig 3. Example of a diatom sample (under light microscope) contaminated with aluminosilicates and sponge spicules.** Sample from the Southern Ocean [17].

samples as low as 0.5 g and 5–20% opal, with 6.5 mg of pure diatom needed for $\delta^{18}O$ and $\delta^{30}Si$ analysis [15]. In contrast, larger raw sediment samples have enabled the recovery of >20 mg pure diatoms and so permitted the analysis of $\delta^{13}C$ [16], which typically requires larger amounts of material. As outlined in the protocol it is also possible, using different sized sieves and/or targeting laminated sediments, to obtain seasonal and/or intra-annual reconstructions [17] or other forms of biogenic silica (e.g., siliceous sponges and radiolaria). Whilst some sediment samples will not be "cleanable" due to the low diatom content, small sample size or inability to remove non-diatom contaminants (Figs 2 and 3), the use of contamination assessment techniques in the protocol allows sample purity to be quantified and ensures that affected samples are not inadvertently used in palaeoenvironmental reconstructions.

## Supporting information

**S1 File. Isotope sample preparation of diatoms for paleoenvironmental research, also available on protocols.io. https://dx.doi.org/10.17504/protocols.io.36wgq4knovk5/v2**. The individual pictured in the S1 File has provided written informed consent (as outlined in PLOS consent form) to publish their image alongside the manuscript.
(PDF)

## Acknowledgments

We thank Teresa Needham and Melanie Leng for their comments on drafts of this protocol and in ensuring the clarity of the steps within it.

## Author Contributions

**Conceptualization:** George E. A. Swann.

**Formal analysis:** George E. A. Swann.

**Funding acquisition:** George E. A. Swann.

**Investigation:** George E. A. Swann, Andrea M. Snelling.

**Methodology:** George E. A. Swann, Andrea M. Snelling.

**Project administration:** George E. A. Swann.

**Supervision:** George E. A. Swann.

**Validation:** George E. A. Swann, Andrea M. Snelling.

**Visualization:** George E. A. Swann.

**Writing – original draft:** George E. A. Swann, Andrea M. Snelling.

**Writing – review & editing:** George E. A. Swann, Andrea M. Snelling.

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
