## [Decision Letter · Decision Letter 0]

26 Dec 2022

PONE-D-22-32814Diatom isotope sample preparation for palaeoenvironmental researchPLOS ONE

Dear Dr. Swann,

Thank you for submitting your manuscript to PLOS ONE. After careful consideration, we feel that it has merit but does not fully meet PLOS ONE’s publication criteria as it currently stands. Therefore, we invite you to submit a revised version of the manuscript that addresses the points raised during the review process.

Please take care about appropriate terminology on 'diatom and isotopes' as suggested by reviewr#1 (Jill Sutton) and note that request for clarification by reviewer#2 is not mandatory. That is, reviewer#2 accepted the protocol as it stands, the comment is for your own convenience. And I agree, this will be an excellent lab guide for specialists.

We look forward to receiving your revised manuscript.

Kind regards,

Alessandro Incarbona

Academic Editor

PLOS ONE

Journal Requirements:

Reviewers' comments:

Reviewer's Responses to Questions

**Comments to the Author**

1. Does the manuscript report a protocol which is of utility to the research community and adds value to the published literature?

Reviewer #1: Yes

Reviewer #2: Yes

2. Has the protocol been described in sufficient detail?

To answer this question, please click the link to protocols.io in the Materials and Methods section of the manuscript (if a link has been provided) or consult the step-by-step protocol in the Supporting Information files.

The step-by-step protocol should contain sufficient detail for another researcher to be able to reproduce all experiments and analyses.

Reviewer #1: Yes

Reviewer #2: Yes

3. Does the protocol describe a validated method?

Reviewer #1: Yes

Reviewer #2: Yes

4. If the manuscript contains new data, have the authors made this data fully available?

Reviewer #1: Yes

Reviewer #2: N/A

**5. Is the article presented in an intelligible fashion and written in standard English?**

Reviewer #1: Yes

Reviewer #2: Yes

6. Review Comments to the Author

Reviewer #1: Dear Authors,

Thank you for sharing this protocol for extracting diatoms from sediments for stable isotope analyses. The protocol is clear and easy to follow. My only criticism is that I would argue that diatoms do not have isotopes, elements have isotopes. Therefore, I have a hard time reading the title of the article: Diatom isotope sample preparation for palaeoenvironmental research. I would suggest that the authors change the title, and subsequently other parts of the manuscript where they use the phrase "diatom isotopes" to: Stable isotope sample preparation of diatoms for paleoenvironmental research." This would make the article more clear. Also, I wonder if you could not be more inclusive to suggest that this type of protocol could also be used for other silicifying organisms, such as sponges. I use a fairly similar method for preparing my diatom, sponge spicule, and radiolarian samples.

Sincerely,

Jill Sutton

Reviewer #2: The article Diatom isotope sample preparation for paleoenvironmental research presents a clear and easy to follow methodology for how to handle samples intended for stable isotopic research. The authors have a step by step process that explains how each stage of the method should work and how many chemical treatments may be necessary at each step.

Among the more important details described in this protocol, are the disaggregation of raw sediment samples, and the removal of organic contaminants. Interestingly, the authors choose to heat the samples to 75C with H2O2 for removal of organic materials, but did not include use of nitric acid in their organic removal methodology. As they have mentioned, their protocol is designed specifically for stable isotope analysis (with the exception of δ15N); I am curious what the author’s reasons are for not choosing this acid to induce a more rapid oxidation reaction (faster than the recommended one week at 75C) with the organic materials. It is not essential, but among modern diatom ecologists, the use nitric acid is common (ex: Trobajo and Mann, 2019; Morales et al., 2013; Romann et al., 2016; Wang et al., 2012; ANSP Protocols for Analysis of NAWQA Algae samples P-13-42), with most reactions being completed in less than 24 hours. Nitric acid also removes carbonate contaminants from sediment samples, so it may be worth adding some detail in the protocols addressing why the longer reaction times with the peroxide reaction are preferred. Again, this is not essential, and most importantly, it does not detract from the protocol’s efficacy in cleaning samples for stable isotopic analysis.

I very much appreciated the detailed descriptions of removal procedures for clay contamination. Aluminosilicates are possibly the most difficult contamination type to remove from samples (even “fresh” samples from planktonic captures), and hold the potential to significantly skew data in such a way as to render it meaningless. The authors lay out an easy to follow process for heavy liquid separation of these contaminating materials, and are very clear about how to effectively and iteratively treat samples to increase purity at each stage.

Lastly, the authors also make a clear point to evaluate samples for purity and potential additional sample preparation prior to isotopic analyses, and they discuss the recovery of heavy liquids used for the sample separations. These steps are crucial for guiding researchers in proper and safe laboratory methodologies, and for informing researchers of the necessary verifications prior to data acquisition.

In total, this protocol draws on established experimental work in diatom stable isotope analytical techniques. It is well informed by the most recent research that demonstrates areas where potential isotopic contamination may occur, and provides clear and explicit directions for users to avoid such errors that could manifest themselves in their data. The steps for the protocol are logical, easy to follow, and provide a check-list for users as they iteratively process their samples. I strongly recommend this protocol for publication.

7. PLOS authors have the option to publish the peer review history of their article (what does this mean?). If published, this will include your full peer review and any attached files.

Reviewer #1: **Yes: **Jill Sutton

Reviewer #2: No

---

## [Author Response · Author response to Decision Letter 0]

24 Jan 2023

Full details are in the response to reviewers document. In summary all requested changes have been made. This includes:

1) changing the title of the manuscript and associated protocol to “Isotope sample preparation of diatoms for paleoenvironmental research”. Text in other relevant sections of the manuscript have also been altered accordingly.

2) Making it clear that the method could also be used on other silicifying organisms.

3) Making it clear that our use of hydrogen peroxide (H2O2) over nitric acid (HNO3) is predominantly one of personal preference.

---

## [Editor Report · Decision Letter 1]

25 Jan 2023

Isotope sample preparation of diatoms for paleoenvironmental research

PONE-D-22-32814R1

Dear Professor Swann,

We’re pleased to inform you that your manuscript has been judged scientifically suitable for publication and will be formally accepted for publication once it meets all outstanding technical requirements.

Kind regards,

Alessandro Incarbona

Academic Editor

PLOS ONE
---

## [Editor Report · Acceptance letter]

8 Feb 2023

PONE-D-22-32814R1 

Isotope sample preparation of diatoms for paleoenvironmental research 

Dear Dr. Swann:

I'm pleased to inform you that your manuscript has been deemed suitable for publication in PLOS ONE. Congratulations! Your manuscript is now with our production department. 

Kind regards, 

on behalf of

Professor Alessandro Incarbona 

Academic Editor

PLOS ONE